# The Long Road from Religious and Ethical Traditions to Welfare of Invertebrates

**DOI:** 10.3390/ani14071005

**Published:** 2024-03-26

**Authors:** Jennifer A. Mather

**Affiliations:** Department of Psychology, University of Lethbridge, Lethbridge, AB T1K 3M4, Canada; mather@uleth.ca

**Keywords:** invertebrates, welfare, ethics, religious values, moral standing, sentience

## Abstract

**Simple Summary:**

The welfare of invertebrates is a result of ethical human behaviour, presently stemming from religious traditions, but how do we get from religious traditions to ethical behaviour to the welfare of invertebrates? The Judaeo–Christian religions are firmly anthropocentric, though they urge utilitarian care for domestic animals. Similarly, the Buddhist tradition of rebirth, possibly as an animal, leads to a belief in ahisma or doing no harm. The theory does not necessarily lead to practice in either case, and Indigenous societies are not usually hunter-gatherers nowadays. Religious practices are often close to good ecology and lead to better consideration. None of these traditions extends much to invertebrates, as only those invertebrates thought ‘worthy’ of consideration are ones that might be sentient. Philosophy is often not based on biological reality, and science demands the simplest possible explanation and ‘objective factual truth.’ The result is a long haul to find and apply the best ethical principles to the welfare of these invertebrates, which are the majority of animals on the planet.

**Abstract:**

Ethical behaviour tends to lead to the welfare consideration of animals, but much less so for invertebrates. Indigenous tradition often valued all animals as having an important role in life on the planet, a practical application of modern ecology. The Judaeo–Christian–Islamic tradition postulated ‘man’ as having dominion over all of Earth, resulting in anthropocentrism and careless practices. In contrast, the Buddhist/Hindu belief in rebirth leads to ahisma, or doing no harm. In the face of capitalist systems, practice does not necessarily follow these beliefs, especially in the ‘shepherding’ of domestic animals. Only Jainist beliefs value the lives of all invertebrates. Philosophers are often divorced from the physiological reality of the animals they muse about, and science’s traditions of objectivity and the simplest possible explanation of behaviour led to ignorance of invertebrates’ abilities. Ninety-seven percent of animals on the planet are invertebrates. We have a long way to go to provide moral standing and welfare consideration, which is consistent with the new information about the sentience of some of these animals.

## 1. Introduction

Ethics must be behind ideas about invertebrate welfare, but it is a long, rocky road to get there. The start has to be some definitions—realising, alas, that they are English and Western. The Cambridge Dictionary defines ethics as “The study of what is morally right or wrong”. The word comes from the Greek ethos, meaning habit, encompassing duties and also consequences. Morality, according to the same source, is “a set of personal or social standards for good or bad behaviour”. McKay and Whitehouse [1] look for the psychological basis of such a set of standards. They see it as biologically derived and evolved, based on a ‘cognitive architecture’ or way of thinking. At the same time, a set of moral values is a culturally produced group of religious and moral representations. They believe that all sets of values look for and judge ‘fairness’ or unbiased evaluation, yet at the same time, all of them use what they call ‘agency detection’, or seeking to find out why things happen and finding different explanations. People tend to evaluate actions not only on the basis of fairness but also in affective terms [2]. Every group begins to see themselves as a whole, forming an ingroup identity and fostering kinship detection. You will pass on your genes to related individuals, so it is important to know and value the related us vs. strangers. This results in the formation of a culture-specific religious/moral identity and a set of rituals that have only social meaning. Think of Jewish prayer shawls, the Navaho dawn song, Christian Sunday, and Buddhist prayer wheels, all getting meaning from and giving meaning to those who perform or use them. Through this group identity, morality becomes all bound up with religion. Singer [3] was explicit about this when he said, “Religious traditions have historically been the principal vehicle by which the status of non-human animals was evaluated”. In action, religious practices are usually the foundation of morality, especially within larger cultural groups [4].

## 2. Religious Traditions about the Welfare of Animals

Since they are foundational, it is useful to explore what different religious traditions have defined and codified as morality with respect to animals. There are several, and their values are different [5]. The Abrahamic traditions, including Islam, are most familiar to us Westerners, so familiar that we are sometimes in danger of seeing them as ‘the’ values. Singer [3] and Szucs et al. [6] point out that this pattern of tradition stresses anthropocentrism, that man is both special and separate from animals, and in the Bible, the book of Genesis explicitly says, “Man shall have dominion over…. every living thing that moveth upon the Earth”. This parallels Aristotle’s idea of the scala naturae, the concept that evolution moved from rocks to plants to ‘lower’ animals and on to humans as the pinnacle of evolution, just below angels. This explicit separation was later fostered in philosophy by Descartes, who believed that animals were mere machines and that we could do anything we liked to, in contrast to humans, who had souls. The Qur’an says, “Then we made you heirs in the land after them” [7]. Such a separation of us from nature persists to this day in Eurocentric societies and has been used to justify the exploitation of land and animals that Western society is based on. Even in science, there is a long history of trying to find a characteristic that sets us apart, disproven one by one. First, there was ‘man the social species’, then ‘man the language user’ (but see bee dances), then ‘man the tool user’, and when that one ran out, ‘man the tool-maker’, until we saw chimpanzees and even New Caledonian crows shaping plant material to make useful tools for getting food and/or water. Such a sought-after separation seems necessary to justify exploitation.

The precepts of Buddhism and Hinduism, the second major set of religious traditions, are quite different. It is fundamental to the Buddhist tradition to do no harm, a principle called ahimsa [8]. This obviously extends to not harming animals, especially as it is believed that when we die, we are reborn in another body, possibly that of a non-human animal. All our actions are submitted to a judging process and are seen to produce and store up either good or bad karma. With accumulated bad karma, you are reborn as a ‘lower’ form of life, likely a non-human animal, with good karma returning as a better-off human, especially as a man. Thus, Buddhists and Hindus have several reasons to treat animals well, especially not to kill them, leading in general to a vegetarian diet. It is not clear that they realised the extent of the animal kingdom, though, or whether they should not harm invertebrates. The Jain sect did so [6], however, and extended their consideration so far as to sweep the path in front of them to avoid hurting insects and avoid eating root vegetables because it would disturb and hurt the small animal fauna of the soil.

The belief systems of ‘aboriginal’ or first inhabitants of many areas of the world, often delineated by differences from settler invaders, are varied but have common themes. Perhaps the best explained in terms of attitudes towards animals are those of North American First Nations [9]. They see power and influence in every part of the ecosystem, even including the rocks. According to the Lakotah Sioux, every animal is important in itself and in its place in the ecosystem, though they did not have that word. Perhaps because they originated as gatherer-hunters, they saw animals as special. They feel a kinship with animals and, in some cases, believe that prey species such as fish were once ‘people’ but transformed into food animals to feed humans (say the West Coast Indians, who explicitly called them to come upriver). Australian Aboriginal groups see the country as a mosaic of clans, territories, and kin groups, identifying strongly with their traditional land and in kinship with particular totem animals, which they never hunt [10]. As gatherer-hunters, aboriginal groups often chase and kill food animals, needing the source of protein, but respect them for ‘giving themselves so the people can eat’. According to the Inuit, if the animals are not respected, they will not make themselves available to be killed. Standing Bear, a Lakotah, speaks of the joy and wonder at the elements and the seasons (something muted by our air conditioners and central heating) and of Earth as mother and Sky as father. Although they were not consciously ecologists, consideration of the welfare of other animal species was almost automatic to these peoples, who lived close to the land.

## 3. From Religion to Practice

To be fair to Western religious traditions, most tell their followers to be kind to the domestic animals under their control, although this is not extended to invertebrates. Care and the killing of domestic animals, is often carefully regulated, and Frayne [11] points out that humans are seen as not so much owners as caretakers of animals. The Jewish kosher dietary tradition is a kind of moral discipline of one’s basic drives, and remember that ingroup formation leads to rituals such as not mixing meat and milk that give ‘meaning’ to what one does. Sharia law likewise dictates halal practices during the keeping of domestic animals as well as their better-known slaughter practices. Deliberately taking the life of an animal is surrounded by rituals in Jewish, Muslim, and First Nations practices to give the animal a swift end and to acknowledge its life as a gift to us; see Grandin and Regenstein’s [12] detailed explanation to ‘meat scientists.’ This utilitarian approach to welfare may naturally accompany domestication, as what is good for the animals produces good meat and other products. In a backward kind of parallel to the drive to be ethical and care for domestic animals, research suggests that people who are abusive to animals as children may abuse other humans in adulthood [13]. These two areas also call attention to the fact that ethical behaviour may be seen to stem from a justice (what is right) and/or a caring (what is kind) approach.

Nevertheless, what the religious traditions dictate may not be what happens in daily life, even though a sample of individuals from a wide variety of countries all said they cared about animals and thought regulation of their welfare was needed [14]. Rollin [15] argues that initially, the husbandry of domestic animals, though utilitarian in focus, resulted in welfare for the animals because care and attention were good for their prosperity. ‘Shepherding’ as protection from predators, an ample supply of appropriate or at least fairly good food, and even rudimentary medical care were all very good for the cows, sheep, or chickens, even though they might end up killed for food. Care measures good for survival may not align with the anima’s larger welfare because they do not fit with the natural history of the species concerned. Industrialisation and the profit motive have changed that [16]. Around 20% of broiler chickens, despite living very short lives, have gait problems that may indicate they are in pain. Remedies such as giving them less crowding, better feed, and longer periods in the dark would result in slower growth and, thus, less profit [17]. The knowledge that raising egg-laying hens in wire cages was poor welfare resulted in a popular concern for them, leading to free movement, including time outdoors [18]. Provision of good welfare is true only for some hens, with ‘marketisation’ of welfare leading to ‘free range eggs’ (and when did you ever see an egg ranging anywhere?) selling for a higher price [19]. Still, it was this lack of regulation that led animal rights activists to campaign, eventually leading to the Five Freedoms, though there are gaps in how welfare legislation is eventually carried out; see [20] for Australia as an example. Conversely, in countries with Buddhist roots, animal welfare is often lacking, though it ‘catches up with us’, so to speak. Really bad welfare practices seen in areas such as the Wuhan ‘wet market’ can be seen as bad karma in the Gita [21], leading to the retribution of COVID-19. Chinese people individually believe that animals ought to be protected, but Chinese laws are lacking [22]. Rahman [7] points out the multiple situations in which fish are suffering in aquaculture in the Middle East. Rollin [15] lists the suffering of cattle in India, where they are theoretically sacred. Even in the subsistence farms of Tibet, some lip service is paid to Buddhist practices [23], but for the most part, animals are raised and slaughtered for food. It is a damning indictment.

## 4. Can Science Partner with Philosophy to Promote Animal Welfare?

Such a partnership sounds logical, yet it does not always succeed. Fraser [24] critiques on the basis of philosophical attitudes, and Webb et al. [25] look more at the scientists. Philosophers may look at general principles rather than details of what specific animal groups can do; the evaluation of invertebrates might be a good example of this. Godfrey-Smith [26] called on the reafference principle to look at the feedback of action paired with copies of movement commands, yet such feedback is not in the cephalopod brain, as control is quite local in the arms of the octopuses he was discussing. Philosophers may lump together unrelated groups, so Mikhalovich and Powell’s [27] emphasis on the welfare of ‘invertebrates’, while an important step forward, could include animals of over 30 phyla, though they acknowledge the problem. The new British regulations spearheaded by the report on invertebrate welfare [28], while ground-breaking, include decapod crustaceans from the Arthropods and Cephalopods from the Molluscs, very different animals who will require very different care and protection [29]. Theorists may not recognise real progress; many welfare-oriented philosophers rail against research practices (‘vivisection’) that are long out of date and end up looking as if they oppose all research.

Still, there are problems when scientists address ethical issues [30]. Scientists are dedicated to objectivity, to facts rather than opinions, and reject ethics and ‘feelings’. Jane Goodall was initially castigated for giving her chimpanzee subjects names because she saw them as individuals. Science has long followed Morgan’s Canon, the belief that the simplest possible explanation of a behavioural phenomenon must be the right one, squashing the possible complexity of motivations even though they might be the correct ones (“It must be reflexes”). Donald Griffin famously ‘came out’ in the 1970s to advocate that we should recognise animals’ subjective experiences [30], but it took a long time for this attitude to permeate studies of animal behaviour. Science is supposed to be objective, so practitioners do not see their own biases; the ‘mirror test’, ref. [31], which was supposed to evaluate animals’ cognitive ability, depends heavily on visual self-identity and grooming, eliminating animals from sentience that do not visually self-evaluate. We are wary of ‘animal welfare activists’, remembering PETA’s destructive anti-research campaign, even freeing captive animals totally unable to live in the wild. Ultimately, regulations that are good for the animals may make research more difficult to carry out; in the US, the IACUCs (Institutional Animal Care and Use Committees) are not loved but tolerated. For researchers in invertebrates, the three Rs (Refine, Reuse, and Replace) of welfare in animal research are regarded warily because Replace is sometimes interpreted as “replace those nice, sensitive mammal subjects with invertebrates to whom you can do anything you like”. In the United States, which has no ethical regulation of invertebrate welfare, invasive and potentially cruel neuroscience research can be carried out. A just-published Japanese investigation of octopus sleep chopped off half of all the arms so that the animals could not pull out the electrodes that were recording brain activity [32] and then commented that it was studying ‘normal’ sleep cycles.

Philosophers and scientists agree, however, on the difficulty of understanding animal consciousness or subjective experience. The Cambridge Dictionary is not much help in defining it, saying it is “the state of being awake, aware of what is around you, and able to think”. The difficulty is that subjective experiences are within oneself and, except for being reported by language, cannot be known outside the individual. Regardless of one’s approach, whether and how to consider animals’ welfare has been based on whether the species in question is sentient [27,29] and thus aware of its present condition and choices for the future. Carruthers [33] evaluates possible explanations for whether animals have Qualia, defined as “instances of subjective conscious experience”, and concludes that no theory explains this very well, though the Global Workspace Theory of Baars [34] comes closest for him. Dawkins [35] takes a more practical approach in our search for phenomenal consciousness in animals as a basis for welfare activity. She evaluates different possible actions to search for consciousness but also notes that it might be better to solve more obvious animal welfare problems. A utilitarian approach may get the animal more benefit, and animals who ‘want’ some environmental condition do not necessarily need consciousness. As we learn more about non-vertebrates, we understand more about their possible sentience and will then need to abandon the present anthropocentrism.

Still, this is an intriguing and relevant problem, and scientists have approached it with many different techniques, measuring physiological responses and behavioural actions. Pain is a particular problem for welfare as it is important to any animal and yet is perceived to include sensory, cognitive, and difficult-to-know affective aspects [36]. Sneddon et al. [37] lay out a series of possible investigations that might evaluate pain in animals, including sensory abilities, connections to the brain, and areas within the brain that process such information. This evaluation is difficult, as physiological responses such as hormone changes may not be the same across diverse animals not related to humans, and aversive behaviour is also likely to be species-typical. Behaviour can at least give us a window into what the animal is ‘thinking’. One promising approach to measuring preference is to give the animal choices, such as Crook’s [38] supply of analgesia to wounded octopuses in a specific location that they previously avoided, although this proves cognition is not affected. Another is to set up a situation where animals are ‘pessimistic’ and see whether that biases their evaluations of ambiguous choices, as when Bateson et al. [39] ‘shook’ bees before giving them a difficult discrimination with an opt-out choice. A third approach is measuring how hard an animal will work to get the condition that it seems to prefer. Still, Dawkins [35] points out that seemingly difficult choices may not even need a conscious decision, so we must be careful not to make sweeping assumptions about awareness.

## 5. What Does All of This Ethical Background Have to Do with Invertebrates

As religious and moral philosophies seldom even recognise that invertebrates existed as animals, they give little background for studies of the application of invertebrate welfare, and even a detailed assessment of what kind of harm we do to animals [40] is aimed only at vertebrates. It is ironic that the careful understanding that many of the aboriginal peoples had of the land and its non-human inhabitants led to their greater consideration of them. Yet their traditions, which often resemble those recommended by modern ecologists, apparently did not also influence us. Perhaps that is where the modern ‘objectivity’ of science, which countered an impression of aboriginal beliefs as subjective and based on ‘stories’, held researchers back.

Philosophers in the Western tradition are still often anthropocentric in their evaluations. They see the link between moral status and ethics; if you harm a being that has moral status, that is morally wrong [41] and unethical. However, this moral status is often human-centred, as Christian tradition says that humans are the only ones ‘made in the image of God’ and all have Formal Moral Status (FMS), regardless of the individual’s cognitive capacity. Animals are assessed for moral status by how they measure up to humans to gain possible status, and sentience and phenomenal consciousness, rather than intelligence, are the criteria [42]. In the face of this, Birch [28] acknowledges the difficulty of knowing and feels that if you cannot specify the ethical priority for a particular animal species, you should invoke the precautionary principle and treat it as well as you can anyway.

Given the difficulty of defining this private subjective experience whose possession should lead to moral standing, a more biological or neural approach should give a firmer basis for judgement. Approaching the problem from a brain perspective, Roth [43] believed in the convergent evolution of intelligence (though not sentience), comparing the mammalian frontal cortex, the insect mushroom bodies, the pallium of birds and fish, and the vertical lobe of cephalopods as parallel higher-order control areas. The neuroscience/cognitive background has led to a greater understanding of these characteristics. The Cambridge Declaration on Consciousness, written by a gathering of neuroscientists and animal behaviourists, was the first to include cephalopods as potentially sentient. Birch et al. [28] used this neural basis for sentience when explicitly advising the British government to set regulations for the welfare of not only cephalopods but also decapod crustaceans. Further on, Pennartz et al. [44] suggest a set of behaviours—qualia richness, situatedness, intentionality, integration, and a balance of dynamic changes and stability—as criteria for consciousness. Assessments of different situations that cephalopod species have reacted to by Mather and Andrade [45] show a similar behavioural array for ‘mind’ in this group. This analysis of invertebrates’ capacity for sentience can be extended to one view: the belief that having a mental life should give an animal moral standing and that cognitive ability, brain size, and capacity to suffer pain should dictate whether we extend moral consideration to invertebrate groups [27].

Some people have separated good welfare from this struggle to gain moral standing. Broom [29] says that “all animal life should be respected”. Dawkins [35] outlines the kinds of things we would like to know about animal consciousness but concludes that making their lives better should come before philosophy. She is also in favour of using animals’ own choices of situations to determine what is best for them as the most valid assessment (and see [38]). With advances in research on some invertebrate groups, we are gaining a better understanding of their behavioural and neural abilities. This has led to better welfare consideration for those being included in the ‘sentience club’ [46], but how do we know what abilities and how much of them should lead to inclusion in ‘the club’? The anthropocentric view of sentience has not led to greater clarity of this designation, and instead, an ethic of care based on each particular animal’s needs is more appropriate. For this inclusive ethic, we can go all the way back to Jeremy Bentham [47], who began the outline of the utilitarian principle as ‘the greatest happiness for the greatest number’, and who wrote in 1789 his oft-quoted sentence: “The question is not “Can they reason?” nor “Can they talk”, but” Can they suffer?”.

## 6. Conclusions

Where does this obviously cursory overview of the journey from religion through ethics to invertebrate welfare take us? First, as pointed out earlier in this paper and in the Editorial, we lack knowledge about invertebrates. This problem is only slowly being solved, but we cannot advance welfare without this understanding. We do not even know what invertebrates live on the planet, as seen earlier in this paper, or how we should behave ethically towards this array of different groups [27]. We need to understand the physiology and behaviour of different species to choose methods to increase welfare and decrease pain. We need to understand the sentience of invertebrates [28] if we are to use it as a benchmark for deciding which animals need special care and consideration. Insects are a particular problem as they represent 60% of all animal species, and we face a huge increase in their use as food without knowledge of their care requirements to ensure their welfare [46]. Here, especially, we see the second problem: misinformation and devaluation due to the anthropocentrism fueled by the Abrahamic religious tradition and human greed. In the face of these two problems, the papers in this collection help inform us about what we are doing to them and what we must start to do about invertebrate animals. Still, it is equally important to, as Birch [28] suggested and Dawkins [35] agreed, invoke the precautionary principle and act ethically towards invertebrates before we know all the details of how they live.

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
