# Peer review of "The Long Road from Religious and Ethical Traditions to Welfare of Invertebrates"

_animals, 2024, doi:10.3390/ani14071005_

Round 1
Reviewer 1 Report (Previous Reviewer 1)
Comments and Suggestions for Authors The article is very interesting and presents a very high substantive level. In my opinion, it does not contain any inaccuracies or unnecessary and unclear fragments. It can and should be published in its current form.
Author Response
thank you very much for your comments
Reviewer 2 Report (Previous Reviewer 2)
Comments and Suggestions for Authors
The revisions made have improved the manuscript compared with the original submission. There are a number of statements that need to be corrected, as detailed below.
Author’s address: ‘Aberta’ should read ‘Alberta’
L. 6: In the first sentence, I suggest to add the word ‘human’ preceding ‘behaviour’.
L. 7: The words ‘it’ and ‘them’ are ambiguous: suggest to replace these by ‘animal welfare’ and ‘these traditions’ respectively.
L. 61: Linking the quote from the biblical book Genesis with ideas of Aristotle is awkward and should be rephrased.
L. 62: The ideas of Aristotle were not linked in any way to ‘evolution’; the suggestion that there is a link between the two is awkward and historically incorrect.
L. 66 – 70: Also here, suggesting a link between the Qur’an and Eurocentrism is a long haul. Rephrase / expand/ or take out.
L. 75: Buddhism and Hinduism are two quite distinct religious traditions; pooling them into the ‘second major religion’, again, is incorrect and awkward. Rephrase.
L. 100: ‘who’ should read ‘whom’.
L. 155: ‘sound’ should read ‘sounds’.
L. 178: ‘animal behaviour’: suggest to change to ‘studies of animal behaviour’.
Author Response
There are a group of minor grammatical corrections, I have made the changes except the one for Line 100, which is correct as written.
Reviewer 3 Report (Previous Reviewer 3)
Comments and Suggestions for Authors
Thank you for the opportunity to re-review this article. Unfortunately, I believe that the previous comments have not been fully addressed in this resubmission.
1) The opinion of the author comes through strongly in the paper, which in places is not necessarily supported.
Unfortunately, I cannot see where the changes have been made to soften the opinion of the writer in text.
2) Some of the sections of the work particularly on pain perception miss some of the most recent and key developments in the field, I would recommend some of the work by Meghan Barrett to update this section, importantly also the debates about the differences between nociception and pain perception are not currently covered.
This paper is still missing some of the key literature emerging currently on insect pain perception, and I would highly recommend even referring to the papers in passing. The author refers to the Elwood paper, however if the author means the 2014 Sneddon and Elwood paper- this is now quite an old paper and there have been many developments in the past decade, including the key work by Dr Birch (I can see one of his papers is cited elsewhere in the paper, however his work on pain and sentience is also important to mention).
3) The coverage of material about religion could be covered in a more nuanced way, for example, some of the rich variety of views within each religious group could be explored in more depth.
I can see that a few more religious examples have been included which is great, however - the work would benefit from a more nuanced approach, even with one case study highlighting the diversity of views within one religion, currently the work reads in a way which suggests that that these enormous religions are united in their views of religion, which is simplistic and misleading.
4) The arguments about how philosophers could improve this area need to include the work which philosophers are already contributing to this area, in particular the philosopher Bob Fischer who has contributed greatly to the field.
I am glad to see that there is one paper now included by Dr Fischer, however despite a page on what philosophers can contribute to this field there is still very little work detailing what current philosophers are publishing on in this field now. There are notable absences of other academics working in this space including Dr Sebo and Professor Drouin. I appreciate the comment about the author not being a philosopher, however if the author is to include a large section of philosophy it is important that work in this area is up to date.
Overall this is an important topic, which due to the sensitivity of the subject matter, needs to be handled with attention to nuance and detail. I do not believe that this paper is a good fit for this journal, however I provide these comments as I hope these will be constructive in developing the work in the future.
Author Response
- No reply is possible.
- I have put in one Barrett paper. Please be aware that this is one paper of a collection of ten. One of the papers will address ethics and insects and another will address insect farming (we, the co-editors, were well aware of the necessity of covering this area), so there is no need to go into detail in this paper.
- There us is not enough time to cover what is a very wide area, of necessity this is cursory.
- Yes, note the addition of the Fischer paper. Again there is not time to go into it in depth, though Jonathan Birch’s contributions are highlighted more.
Reviewer 4 Report (New Reviewer)
Comments and Suggestions for Authors
This is a thought-provoking review about the influence of ethics, morality and religious traditions on animal welfare and specifically invertebrates. It discusses the expansion of animal protection from religious traditions via scientific understanding of cognitive ability, behaviour, sentience and pain. The difficulty of defining private subjective experiences is an argument for extending moral consideration to invertebrates. The author is passionate about the topic and my main suggestions are about providing greater balance to some parts of the discussion.
The section beginning on Line 109 entitled ‘From religion to practice’ discusses animal welfare legislation. This section should acknowledge specific existing legislation which protects invertebrates. For example, the most commonly included invertebrates are the cephalopods (e.g. octopus, squid). Legislation in New Zealand also covers crabs, lobsters and crayfish (Animal Welfare Act. 1999 https://www.legislation.govt.nz/act/public/1999/0142/latest/whole.html), while Norway includes all decapods as well as honey bees (Animal Welfare Act. (2010). https://www.regjeringen.no/en/dokumenter/animal-welfare-act/id571188/).
Lines 188-190 state ‘In the United States, which has no ethical regulation of invertebrate welfare, invasive and potentially cruel neuroscience research can be carried out.’ It should be noted there is growing awareness of invertebrate welfare because on 7 September 2023, the US National Institutes of Health (NIH) asked for feedback on proposed guidelines that, for the first time in the United States, would require research projects involving cephalopods to be approved by an ethics board before receiving federal funding (https://www.nature.com/articles/d41586-023-02887-w).
Lines 129-134 argue that utilitarian animal husbandry practices brought incidental animal welfare benefits but industrialization and the profit motive have changed that. There should be some recognition that community and industry have differing definitions of good animal welfare which are satisfied by different metrics. Community prioritises anthropomorphic concepts of cleanliness and freedom (free range). Industry procedures are more science-based and aimed at protecting animals from disease (biosecurity), predators and aggression of conspecifics (animals belonging to the same species). For example, lameness and aggression may be more prevalent in sows housed in groups rather than individually (Maes, D., Pluym, L. & Peltoniemi, O. Impact of group housing of pregnant sows on health. Porc Health Manag 2, 17 (2016). https://doi.org/10.1186/s40813-016-0032-3). Free-range poultry are more susceptible to infection with avian influenza (AI) spread by wild birds. The environmental and economic cost of destroying hundreds of thousands of commercial poultry during outbreaks of AI may overwhelm any perceived or real animal welfare benefits of free-range systems (Scott A, Hernandez-Jover M, Groves P, Toribio JA. An overview of avian influenza in the context of the Australian commercial poultry industry. One Health. 2020 May 11;10:100139. doi: 10.1016/j.onehlt.2020.100139. PMID: 32490131; PMCID: PMC7256052.
https://www.abc.net.au/news/2020-09-18/victorian-bird-flu-outbreak-raises-concerns-free-range-farming/12669544)
Some minor points;
The simple summary commences with this question – ‘Welfare of invertebrates is a result of ethical behaviour, presently stemming from religious traditions, but how do we get to it from them?’ It may be clearer to reword this to ‘How do we get from religious traditions to ethical behaviour to welfare of invertebrates?’
This is the final sentence of the abstract – ‘With the new information about sentience in some of these animals, moral standing and welfare consideration has sometimes been extended to invertebrates, but we have a long distance to go to consider recognition and care of all these 97% of the animals on the planet.’ It may be clearer to reword this to ‘Ninety-seven percent of the animals on the planet are invertebrates. We have a long distance to go to provide moral standing and welfare consideration which is consistent with the new information about sentience in some of these animals.’
Author Response
- For the section beginning on Line 100, note that this was covered in the Editorial so I did not duplicate it here.
- For the section starting at line 188. I am aware of the effort to convince NIH that cephalopods should be protected by welfare legislation as I am a member of the group that has been trying to persuade them. When I originally wrote this paper, they had rejected inquiries from congressional members after our presentation to them. Though they have now asked for submissions for feedback for proposed legislation, it is only advisory and as such our group has suggested stronger wording to make it a requirement. I hope this will come to pass but am waiting to see.
- For lines 129, this is an important point and I have added a sentence emphasizing it.
- I have made the changes to the Summary and Abstract to make them clearer.
This manuscript is a resubmission of an earlier submission. The following is a list of the peer review reports and author responses from that submission.
Round 1
Reviewer 1 Report
Comments and Suggestions for Authors
The text is convincing, wise and necessary. The argument is perfectly supported by world literature. I consider the passages concerning the selective treatment of animals by scientists to be particularly important. This is another important voice in the matter of getting rid of irrational prejudices against invertebrates in science. I did not find any questionable or redundant passages in the article. In my opinion, however, the article lacks two important threads, so I posted two comments that, I think, are worth considering:
1. In the author's argument, I miss a clear reference that in the modern world economic factors have become at least as important, or even more important than religion. Since the Covid pandemic, e.g. materials from the "wet animal market" in Wuhan have gained popularity in the media. The horrific ways in which animals (mammals, birds, fish, reptiles, cephalopods, crustaceans) have been treated by sellers and buyers on a massive scale for many years in many such places in Asia (in societies with Buddhist and Confucian religious traditions) and beyond, lead to the conclusion that individual desire for profit is as important in these matters as religion and any moral norms. Yes, the basis of human indifference to the suffering of animals is the assimilation of ethical norms in childhood, usually of religious origin, but the direct cause of animal suffering on a large scale is the desire to earn money, which has nothing to do with ethics or religion. Such business projects as, for example, large octopus farms for meat (Italy), mass farms of consumable insects for millions of individuals (all of Europe), 26-storey piggeries, "producing" 1.2 million pigs per year (China), oceanic fish aquaculture with an area of 650 km2 (Brazil) are developed for profit in countries with different religious traditions, but without a direct connection to religion. These are issues worth at least mentioning in the text (e.g. after verse 141?).
2. Invertebrate welfare issues are completely overlooked in applied sciences related to agriculture (with notable exceptions in recent years regarding the effects of insecticide use on pollinating insects), forestry, biological monitoring. As part of "pest control", environmental quality assessment, and biodiversity monitoring etc. huge numbers of invertebrates are cruelly killed, much greater than in laboratory studies. Reducing the disregard for invertebrate welfare during such activities requires a completely different approach than in the case of laboratory research, but it is equally important (after line 180, I suggest?).
Reviewer 2 Report
Comments and Suggestions for Authors
This brief review paper addresses a topic that receives increasing attention concurrent with the rapid growth of industrial-scale insect production of insects for food and and feed. Overall it reads well, although terminology is used rather loosely at places; see comments below.
The main point for improvement is to clarify the main message the paper is aiming to convey. As I perceive it, this message is: 'certain non-western cultures / religions apply higher ethical standards for treating invertebrate animals.'
L. 7: Throughout the text, the author refers to ‘Judaeo-Christian religions’. It is important to stress that it would more accurate to refer to ‘Abrahamic religions’, which also includes Islam (Ibrāhīm in the Quran; which is equally anthropocentric; as acknowledged in line 54), as all three religions worship ‘the God of Abraham’. Moreover, the term ‘religion’ is most adequately ascribed to these Abrahamic traditions, as the term religiō is most likely derived from religare, meaning re-connect (i.e. between a (chosen) creature and its Creator). Such a devotional relationship with and blind obedience to a divine creator is absent in most non-Abrahamic traditions.
L. 8: It is unclear what is meant with “indigenous religious practices”, as most non-Abrahamic traditions are not religious in the sense described above. Moreover, it could be argued that Judaism, Christianity and Islam are all endogenous to the Western world, as they emerged in the Near- and Middle East, not Europe. It is widely accepted that Indo-European spirituality (which also includes Vedic traditions!), e.g. varieties of Nordic, Slavic, Germanic, Roman and Greek polytheism, also “often valued all animals having an important role in the life on the planet, much like all modern ecology” (line 17). However, although these Indo-European traditions can be called ‘ecocentric’, their practices did involve animal sacrifice for example, in rituals that were thought to increase fertility of plants, animals and humans themselves.
L. 10: ‘deep ecology’; this term needs a definition/explanation; I expect it to be unfamiliar to the readership of ‘Animals’.
Line 32: To define ethics as derived from ethos is appropriate, for example, for most Indo-European traditions, most notably Greco-Roman traditions (e.g. in accordance with the works of Plato and Aristotle). Both authors, in particular Aristotle, acknowledges the role of the group in which it produced, e.g. civilization and proper education, which, in a way, turns it into a particularistic doctrine. This definition of ethics, however, does not match with the most dominant ethical theories of today, i.e. utilitarianism and deontology. Both of these modern theories have a certain pretense of universality, i.e. their validity is said to be based on universal principles and thus independent of upbringing and kinship. The most notable example of this (“western-centric”) kind of philosophizing is formalized in the Universal Declaration of Human Rights, which is based on the reasoning that incorporates both duties (deontological principles) and consequences (utilitarian calculation).
L. 38: A definition of ‘fairness’ and an explanation of how it relates to ethics/morality would help the reader to understand its meaning in the context of this paper.
L. 61: Descartes did not maintain that animals were things, in the way that a rock can also be considered a thing. According to Descartes, who was gripped by the mechanistic understanding of the universe following the Copernican Revolution, animals were machines (automata).
L. 65: Explain what ‘Eurocentric societies’ are and how this concept relates to morality with respect to animal welfare. Is this term necessary for the story line?
L. 65-71: Unclear where the reasoning leads to since it suddenly ends. It seems that the implicit conclusion is that man and animals are not fundamentally different? What consequence(s) does this conclusion have for ethical behaviour vis-a-vis animals?
L. 72: As mentioned above, it is not entirely adequate to call Buddhism a religion. Moreover, ashima is part of ‘right conduct’ (sīla). Wrong conduct should be understood as a “fault” (not a kind of “sin”, as in Abrahamic tradition); to err is not “evil” in the Vedic traditions. To harm an animal is thus a sign of stupidity, foolishness, or harshness; this is a matter of knowledge, not a matter of morals (being “good or bad”). According to the doctrine of karma, the one who is at fault will simply have to endure the consequences of his own misconduct and, most importantly, become determined not to err again. This is in stark contrast with the prospect of (“divine”) punishment for an “immoral act” (i.e. the risk of eternal damnation because one violated the moral law revealed through the word of the one and only God).
L. 88: The indigenous populations of the (North-)American continent have often been depicted as a “wild ecologists”, following the notion of “noble savages” as used by philosophers like Jean-Jacques Rousseau. Despite the fact that these peoples indeed viewed nature and all living things as a sacred whole, and did not hunt specific animals, it seems like a kind of romanticization, as they lived mainly by hunting.
L. 93: When we are concerned with invertebrates, this example is also somewhat strange to me: Australian Aboriginals have a rich history of consuming insects (referred to as “bush tucker”), which includes honey ants and the larvae of certain moths (so-called “witchetty grubs”).
L. 107: Shechita (Jewish ritual slaughter) is not to be confused with kashrut (Jewish dietary laws). Moreover, shechita specifically requires that animals are slaughtered without being stunned or sedated (i.e. whilst still sentient and able to feel pain), specifically by cutting the animal's throat (severing the trachea and esophagus). Hence this kind of rituals are often considered problematic from the perspective of animal welfare.
L. 110: Unsurprisingly, given the religious root of both Judaism and Islam, similar considerations apply to the Islamic prescription of dhabīḥah (“halal slaughter”). While cutting the throat, damage to the central nervous system must be avoided, as the procedure (or ritual) aims for exsanguination, i.e. the animal must bleed to death and thus should not die beforehand.
L. 135-141: This is easily explained by the fact that most of these countries are heavily modernized through colonization and ongoing globalization, and, subsequently, many actors in these countries have adopted the “western modus operandi”, i.e. engage in activities that are very similar to the industrial-technological-capitalist complex as we know it.
L. 181-264: Note that only deontological and utilitarian doctrines rely on the concept of “moral status”, and thus require evidence for sentience (or “phenomenal consciousness”). Given that both these modern theories are secularized variants of Abrahamic religions (see the works of Friedrich Nietzsche, or Anscombe (1958)), it seems that many animal welfare scientist and philosophers are still “in search for the soul”. Note in addition that (Neo-)Aristotelian ethics and philosophy allows us to think about the ethical treatment of animals and animal welfare without these (arguably confused) concepts. For example, see Hacker (2002) on the nonsense that arises when we assume the common concept of subjectivity, Bennett and Hacker (2022) on why consciousness is not “in the brain”, and Hursthouse (2011) on why we would be better off without the concept of moral status.
L. 188: ‘explanation’ should read ‘explanations’
L. 205-206: This statement, of anthropomorphic signature, touches upon a central issue of the topic and deserves to be elaborated on: what criteria do scientists have available to derive information on subjective experience / sentience of an (invertebrate) animal from? Are behavioral criteria more informative than physiological / neurobiological measurements? This is not limited to preference behavior; however, the following sentences are limited to this. Could the author provide other examples?
L. 219-220: The statement on resemblance of belief systems of aboriginal people and belief systems of contemporary ecologists (if any) is far-fetched and unconvincing. Ecology is a biological discipline that collects factual, numerical information from natural systems in an attempt to understand their functioning and shows in my opinion no resemblance to any kind of belief system.
L. 215-264: The last section of the paper is lacking a much-needed perspective on the future of how animal ethicists could contribute to better welfare of invertebrate animals. The author is invited to add such an outlook: is the current state of affairs in animal ethics, rooted as the author argues in different religious traditions, applicable to invertebrates as such or are modifications needed? Or is a completely new framework that leaves behind any religious tradition necessary? Is differentiation between invertebrates, comprising 30 animal phyla, called for? Questions like these may be put on stage to sketch perspectives on the topic of this review.
Literature cited:
Anscombe, G. E. M. (1958). Modern moral philosophy. Philosophy, 33(124), 1-19.
Bennett, M. R., & Hacker, P. M. S. (2022). Philosophical Foundations of Neuroscience. John Wiley & Sons.
Hacker, P. M. (2002). Is there anything it is like to be a bat? Philosophy, 77(2), 157-174.
Hursthouse, R. (2011). Virtue ethics and the treatment of animals. In T. L. Beauchamp & R. G. Frey (Eds.), The Oxford Handbook of Animal Ethics (pp. 119-143).
Comments on the Quality of English Language
L. 5: ‘Aberta’ should read ‘Alberta’.
L. 122: ‘animal’ should read ‘animals’
L. 134: ‘caried’ should read ‘carried’
L. 143: ‘sound’ should read ‘sounds’
L. 144: ‘looks’ should read ‘look’
L. 168: ‘animal’ should read ‘animals’
Reviewer 3 Report
Comments and Suggestions for Authors
Thank you for the opportunity to review this paper on the long road from religious and ethical traditions to the welfare of invertebrates. This paper raises some interesting points however I believe that this piece of work is not a good fit for this journal, and may be better placed in a philosophy journal.
I had several concerns about this paper:
1) The opinion of the author comes through strongly in the paper, which in places is not necessarily supported by the literature, more reference to the literature would strengthen this piece of work.
2) Some of the sections of the work particularly on pain perception miss some of the most recent and key developments in the field, I would recommend some of the work by Meghan Barrett to update this section, importantly also the debates about the differences between nociception and pain perception are not currently covered.
3) The coverage of material about religion could be covered in a more nuanced way, for example, some of the rich variety of views within each religious group could be explored in more depth.
4) The arguments about how philosophers could improve this area need to include the work which philosophers are already contributing to this area, in particular the philosopher Bob Fischer who has contributed greatly to the field.
Overall, while this paper raises interesting points this piece of work is not a good fit for this journal.
Comments on the Quality of English LanguageThe standard of writing is good, however the language is less formal than would be expected for a journal of this type.